# Genome-Wide Analysis and Expression Profiles of the VOZ Gene Family in Quinoa (*Chenopodium quinoa*)

**DOI:** 10.3390/genes13101695

**Published:** 2022-09-21

**Authors:** Pibiao Shi, Runzhi Jiang, Bin Li, Deling Wang, Di Fang, Min Yin, Mingming Yin, Minfeng Gu

**Affiliations:** Xinyang Agricultural Experiment Station of Yancheng City, Jiangsu Academy of Agricultural Sciences, Yancheng 224049, China

**Keywords:** *Chenopodium quinoa*, VOZ transcription factor, genome-wide analysis, abiotic stress, gene expression

## Abstract

Vascular plant one zinc-finger (VOZ) proteins are a plant-specific transcription factor family and play important roles in plant development and stress responses. However, little is known about the *VOZ* genes in quinoa. In the present study, a genome-wide investigation of the VOZ gene family in quinoa was performed, including gene structures, conserved motifs, phylogeny, and expression profiles. A total of four quinoa *VOZ* genes distributed on three chromosomes were identified. Based on phylogenetic analysis, *CqVOZ1* and *CqVOZ3* belong to subfamily II, and *CqVOZ2* and *CqVOZ4* belong to subfamily III. Furthermore, the VOZ transcription factors of quinoa and sugarbeet were more closely related than other species. Except for *CqVOZ3*, all the other three *CqVOZs* have four exons and four introns. Analysis of conserved motifs indicated that each *CqVOZ* member contained seven common motifs. Multiple sequence alignment showed that the *CqVOZ* genes were highly conserved with consensus sequences, which might be plausibly significant for the preservation of structural integrity of the family proteins. Tissue expression analysis revealed that four *CqVOZ* genes were highly expressed in inflorescence and relatively low in leaves and stems, suggesting that these genes had obvious tissue expression specificity. The expression profiles of the quinoa *CqVOZs* under various abiotic stresses demonstrated that these genes were differentially induced by cold stress, salt stress, and drought stress. The transcript level of *CqVOZ1* and *CqVOZ4* were down-regulated by salt stress and drought stress, while *CqVOZ2* and *CqVOZ3* were up-regulated by cold, salt, and drought stress, which could be used as abiotic stress resistance candidate genes. This study systematically identifies the *CqVOZ* genes at the genome-wide level, contributing to a better understanding of the quinoa VOZ transcription factor family and laying a foundation for further exploring the molecular mechanism of development and stress resistance of quinoa.

## 1. Introduction

Abiotic and biotic stresses can cause serious harm to plant growth and development, resulting in lower crop yield and quality [1,2]. In order to survive better, plants have evolved a series of molecular response mechanisms to adapt to various stresses, and a large number of related genes have been mined and studied [3,4]. As key molecular switches regulating stress-responsive gene expression, transcription factors play vital roles in plant response to various abiotic stresses and are recognized as promising candidates for genetic improvement [5,6,7]. Currently, more than 60 transcription factor families have been found in plants [8], some of which, such as WRKY, MYB, NAC, bHLH, and bZIP, were identified to be involved in mediating abiotic stress responses in different species [9,10,11,12,13].

One of the plant-specific transcription factor families, vascular plant one zinc-finger (VOZ) transcription factors are exclusively present in many higher plants, including vascular plant and *Physcomitrella patens* [14]. *VOZ* is a multifunctional gene that regulates a variety of biological processes, such as flower induction and development, pathogen defense, and various abiotic stress responses [15]. This family was first identified and characterized in *Arabidopsis* with two members, *AtVOZ1* and *AtVOZ2*, which bind to GCGTNx7ACGC palindromic sequence in the promoter region of V-PPase gene (*AVP1*) in vitro [14]. VOZ proteins have two conserved domains, A and B. The B domain, also named the VOZ domain, contains a zinc finger motif and a basic region. Further study showed that the VOZ domain might have the dimerization function of the DNA domain in all VOZ proteins [16]. In *Arabidopsis*, *AtVOZ1* was especially expressed in phloem, while *AtVOZ2* was more abundant in roots [14]. The *GmVOZs* were found to express in roots and leaves of soybean at the seedling stage [17]. Recently, it has been reported that AtVOZs interacted with phytochrome B to regulate the transition from vegetative growth to flowering by controlling the expression of *FLOWERING LOCUS T* (*FT*) and *FLOWERING LOCUS C* (*FLC*), and *atvoz1-1 atvoz2-1* double mutant exhibited delayed flowering time [18,19]. On the other hand, another study showed that VOZs modulate the function of *CONSTANS* (*CO*) to promote flowering, independent of *FLC* [20]. The rice *osvoz1 osvoz2* double mutant displayed severe dwarfism and cell death, and *OsVOZ1* acted as a negative regulator and *OsVOZ2* functioned as a positive regulator to resist fungal pathogen *Magnaporthe oryzae* [21]. Studies have also shown that VOZ can respond to biotic and abiotic stresses. For instance, overexpression of *AtVOZ2* induced the transgenic plants to be more susceptible to drought and cold stress but enhanced the resistance against a fungal pathogen, *Colletoricum higginsianum* [22]. The *voz1 voz2* double mutant showed increased tolerance to drought and cold stress but decreased resistance to heat stress and pathogens in *Arabidopsis* [23,24]. It was demonstrated that *AtVOZ1* and *AtVOZ2* suppressed the expression of *DEHYDRATION-RESPONSIVE ELEMENT BINDING FACTOR 2C* (*DREB2C*) and *DREB2A*, respectively, to respond to heat stress negatively [25,26]. *AtVOZ1* and *AtVOZ2* could also enhance the salt tolerance by directly or indirectly regulating the transcription level of several salt-responsive genes [16]. Based on soybean genome information, six *GmVOZs* were identified [27], and then they were cloned and characterized for further study [15]. Overexpression of *GmVOZ1G* in soybean hairy roots enhanced tolerance to drought and salt stress, while RNA interference plants showed more sensitivity to these stresses [15]. These results suggest that VOZ transcription factors play crucial roles in the regulation of plant growth and development, and have important application values in the genetic improvement of stress resistance of crops. Therefore, identifying and characterizing new *VOZ* genes from multiple plant species provides a reliable view to understanding this highly conserved family.

Quinoa (*C. quinoa* Willd.), an annual dicotyledonous herbaceous plant, originated from the Andean region of South America and has been cultivated for approximately 7000 years [28]. It belongs to the Amaranthaceae family, which comprises hundreds of species, including some representative crops such as sugar beet (*Beta vulgaris*), spinach (*Spinacia oleracea*), and purslane (*Portulaca oleracea*) [29,30]. Quinoa has been regarded as a pseudo-cereal because of its grain characteristics [31]. Due to remarkable nutritional properties, abundant proteins, balanced essential amino acids, enriched vitamins, significant amounts of minerals, dietary fiber, unsaturated fatty acids, and absence of gluten, quinoa has been called “golden grain” and “superfood” by the international nutritionists [32,33]. Meanwhile, it exhibits strong resistance to various climatic and soil conditions such as cold, high salinity, and drought, which allows quinoa to be cultivated in marginal lands [34,35,36]. However, abiotic stresses can also affect the growth and development of quinoa. Cold damage affects the quinoa final yield by influencing the nutrient uptake, prolonging the seedling period, and leading to dwarfing of vegetative organs at the seedling stage [37]. Among different growth stages of quinoa plants, such as the establishment, flowering, and seed filling stages, the salinity sensitivity of seedlings is higher than that of mature plants [38]. As a major crop for global food security and sustainability under complex and changeable environmental conditions, quinoa has the potential to become the new staple food. The year 2013 was declared the International Year of Quinoa by the United Nations Food and Agricultural Organization (FAO), which meant the potential of this emerging crop was increasingly recognized worldwide [39]. In recent decades, quinoa has attracted extensive attention, especially from researchers and consumers. With the publication of a high-quality reference genome, the research on functional genomics of quinoa has been greatly promoted [40]. *VOZ*, a multifunctional gene regulating plant growth and development and abiotic stress responses, has been identified and studied in many model plants such as Arabidopsis, rice, and soybean. However, a genome-wide investigation of the VOZ family has not yet been performed in quinoa. In the present study, four putative *VOZ* genes were identified in quinoa. Moreover, basic physical and chemical properties, chromosome distribution, phylogeny, gene structure, conserved motifs, and the expression profiles of *CqVOZ* gene family were systematically analyzed, laying a foundation for further investigation of their functions. The results provide insights into understanding the molecular mechanisms of development and stress tolerance in quinoa and other crops.

## 2. Materials and Methods

### 2.1. Plant Materials and Stress Treatments

SL1 (Suli No. 1 from Jiangsu Academy of Agricultural Sciences) was used as the material throughout the study. Plants were grown in the greenhouse for tissue expression analysis. Mature roots, stems, and leaves were collected from the same plants. Inflorescence was sampled at the flowering time. Seeds were harvested at the 10, 20, and 30 days after pollination (DAP). For stress treatments, seeds were surface-sterilized with 3% H_2_O_2_ for 1 min, rinsed several times using sterile water, and germinated at 28 °C under long-day conditions with a photoperiod of 16 h light/8 h dark in a phytotron. Seven days later, the uniform seedlings were transferred to half-strength Hoagland nutrient solution until the sixth true leaf stage. And then the seedlings were treated with 300 mM NaCl, 20% PEG 6000, and 4 °C low temperature, respectively. The root samples were collected at 0, 3, 6, 12, and 24 h after abiotic stress treatments. All the samples mentioned above were immediately frozen in liquid nitrogen and stored at −80 °C for RNA extraction. Three biological replicates were employed for each sample. 

### 2.2. Genome-Wide Identification and Chromosomal Location of the VOZ Genes in Quinoa

The sequences of two Arabidopsis AtVOZ proteins were retrieved from the TAIR database (http://www.arabidopsis.org, accessed on 3 July 2022) and then used as queries to identify *VOZ* genes in the quinoa v1.0 genome database (https://phytozome-next.jgi.doe.gov/info/Cquinoa_v1_0, accessed on 3 July 2022). The local BLASTP program with the expectation value (E) < 1 × 10^−5^ was employed in the database search. The number of amino acids (AA), isoelectric point (pI), and molecular weight (MW) of the *CqVOZs* were computed by the pI/MW tool from ExPasy (http://au.expasy.org/tool.html, accessed on 4 July 2022). Subcellular localization prediction was performed in PSORT WWW Sever (https://psort.hgc.jp/, accessed on 12 July 2022). Based on the chromosomal positions of the *CqVOZs* searched from ChenopodiumDB (https://www.cbrc.kaust.edu.sa/chenopodiumdb/about.html, accessed on 6 July 2022), their chromosomal distributions were drawn using MapChart software [41]. 

### 2.3. Phylogenetic Analysis

Multiple sequence alignments of the full-length sequences of VOZ proteins from *C. quinoa* (Cq), *Arabidopsis thaliana* (At), *Gossypium hirsutum* (Gh), *Oryza sativa* (Os), *Sorghum bicolor* (Sb), *Zea mays* (GRMZM), *Glycine max* (Gm), and *B. vulgaris* (Bv) were performed by using ClustalX 2.0 software with the default settings as described earlier [42]. An unrooted phylogenetic tree was constructed through the Neighbor-Joining (NJ) method using MEGA 5.0 software with the following parameters: pairwise alignment, Poisson correction model, uniform substitution rates, complete deletion, and 1000 bootstrap replicates. 

### 2.4. Gene Structure and Conserved Motif Analysis

Gene exon-intron structure characteristics were analyzed by alignment of the open reading frames (ORFs) with their genomic DNA sequences and visualized using the Gene Structure Display Server (GSDS 2.0, http://gsds.gao-lab.org/, accessed on 14 July 2022). The conserved motifs of the CqVOZ proteins were identified in Multiple Em for Motif Elicitation (MEME) program (https://meme-suite.org/meme/tools/meme, accessed on 14 July 2022) with the following parameters: The number of repetitions is any, the maximum number of motifs is 10, and the optimum motif width is between 6 and 50. CqVOZ protein sequences were submitted to Pfam (http://pfam.xfam.org, accessed on 3 July 2022) to identify conserved VOZ domains. In addition, amino acid sequence alignment of the VOZ domains containing several conserved motifs was performed by using ClustalX 2.0.

### 2.5. Protein Structure Analysis of CqVOZs

The secondary structures of CqVOZ proteins were predicted using the GOR IV tool (https://npsa-prabi.ibcp.fr/cgi-bin/npsa_automat.pl?page=npsa_gor4.html, accessed on 4 July 2022) according to the default parameters. The three-dimensional (3D) structures of CqVOZ proteins were scanned on the SWISS MODEL program (https://swissmodel.expasy.org/, accessed on 4 July 2022).

### 2.6. RNA Extraction and qRT-PCR Analysis

Total RNA was extracted from each sample using the RNeasy Plant Mini Kit (Qiagen, Germany) and RNase-Free DNase Set (Qiagen, Germany) as per the manufacturer’s instructions. RNA concentration and quality were determined by a Thermo 2000 Bioanalyzer with an RNA NanoDrop (Thermo Fisher Scientific, Waltham, MA, USA). The first strand cDNA was synthesized by using SuperScript™III First-Strand Synthesis SuperMix with gDNA Eraser (Thermo Fisher Scientific, USA) according to manufacturer’s recommendations. To identify the relative expression level of the *CqVOZ* gene family in different tissues and under different abiotic stress treatments, quantitative real-time polymerase chain reaction (qRT-PCR) was carried out using Power SYBR^®^ Green PCR Master Mix (Applied Biosystems, USA). The quinoa Elongation Factor 1 α (*EF1α*) gene was used as an internal control. All the primers used for qRT-PCR were designed using Primer Premier 6.0 and Beacon Designer 7.8 software, and the sequences are shown in Table 1. qRT-PCR was conducted in a total volume of 20 μL, containing 1 μL cDNA, 10 μL Power SYBR^®^ Green Master Mix, 8 μL ddH_2_O, 0.5 μL forward primer (10 μM) and 0.5 μL reverse primer (10 μM), using the CFX384 Connection Real-Time System (Bio-Rad, USA). The qRT-PCR program was as follows: 95 °C for 1 min, then 40 cycles of 95 °C for 15 s, 63 °C, for 25 s. Three biological repeats were performed for each sample. The relative expression levels of *CqVOZs* were analyzed by the 2^−∆∆CT^ method [43]. The gene expression heatmap of *CqVOZs* in different quinoa tissues was drawn by HemI 1.0 software [44]. The expression profiles of *CqVOZs* under different abiotic stress treatments were drawn by Origin 8 software.

## 3. Results

### 3.1. Identification and Characterization of CqVOZ Gene Family

To identify *VOZ* genes in quinoa, the sequences of AtVOZ proteins of Arabidopsis were employed as queries to screen the quinoa genome through BLASTP. A total of four VOZ transcription factors were obtained and renamed from *CqVOZ1* to *CqVOZ4* based on their coordinate order on quinoa chromosomes (Chr) (Table 2). The basic information of *CqVOZs* including chromosomal location, open reading frame (ORF) length, amino acid (AA) number, molecular weight (MW), and isoelectric point (pI), was also analyzed and shown in Table 2. The ORF length of the *CqVOZs* ranged from 1230 bp (*CqVOZ3*) to 1569 bp (*CqVOZ2*), with an average of 1455 bp. The AA number was between 409 to 522, correspondingly. The MW of the CqVOZ proteins varied from 45.93 kDa (CqVOZ3) to 58.08 kDa (CqVOZ2), with an average of 54.06 kDa. The pI values ranged from 5.48 to 7.56, among which CqVOZ1, CqVOZ2, and CqVOZ4 were acidic proteins (pI < 7), while CqVOZ3 was a basic protein (pI > 7). Subcellular localization analysis demonstrated that all four *CqVOZs* were mainly located in the nucleus. Moreover, to examine the genome distribution of the CqVOZs, chromosomal mapping was plotted using MapChart software. The results showed that the *CqVOZs* were unevenly distributed across three quinoa chromosomes, two genes (*CqVOZ1* and *CqVOZ2*) were located on chromosome 1, while *CqVOZ3* and *CqVOZ4* were located on chromosomes 2 and 4, respectively (Figure 1).

### 3.2. Phylogenetic Analysis of CqVOZ Genes

In order to better understand the phylogenetic relationship of the *CqVOZ* gene family in quinoa, an unrooted phylogenetic tree was constructed using the Neighbor-Joining grouping method based on multiple sequence alignment of VOZ proteins of quinoa (four *CqVOZs*), *Arabidopsis* (two *AtVOZs*), rice (two *OsVOZs*), cotton (seven *GhVOZs*), sorghum (two *SbVOZs*), corn (five *GRMZMVOZs*), soybean (six *GmVOZs*) and sugarbeet (two *BvVOZs*) (Appendix A). As shown in Figure 2, all 30 *VOZs* were classified into four groups, designated Group I to IV: 10 members belonged to Group I, 3 to Group II, 8 to Group III, and 9 to Group IV. However, four *CqVOZs* existed only in Group II and III. *CqVOZ1*, *CqVOZ3,* and *Bv8_182380_cpjk.t1* were on the same evolutionary tree branch, while *CqVOZ2*, *CqVOZ4,* and *Bv9_209290_ninc.t1* were on the same branch, suggesting that the *VOZs* of plants in the same family had the highest homology and were most closely related. Furthermore, *CqVOZ2*, *CqVOZ4,* and *AtVOZ2* belonged to the same group, implying the two *CqVOZs* might have similar biological functions with *AtVOZ2* as previously reported [22,26]. Interestingly, we found that all the VOZs of rice, sorghum, and corn consisted of Group IV, while the *VOZs* of quinoa, *Arabidopsis*, cotton, soybean, and sugarbeet were unevenly distributed in the other three groups. This finding revealed that the VOZ gene family diversified after the divergence of monocotyledons and dicotyledons. 

### 3.3. Gene Structures and Conserved Motifs of CqVOZs

Differences in exon-intron structure within families play a critical role in the evolution of multiple gene families [45]. To understand the diversity of *CqVOZs*, the Gene Structure Display Server (GSDS) analysis was conducted using their genomic sequence and corresponding cDNA sequence. As shown in Figure 3, the genome sequence lengths of *CqVOZs* ranged from 4861 to 6303 bp, and the lengths of CDS varied from 1230 to 1569 bp. All the members, except *CqVOZ3*, have similar gene structures. They contain two 5′ untranslated regions (UTRs), four exons, and one 3′ UTR, while *CqVOZ3* possesses three exons and one 3′ UTR. The gain or loss of exons implies that the function of these genes may be altered. 

The full-length protein sequences of four CqVOZs were analyzed using the MEME program to identify their conserved motifs. These motifs were marked as Motif 1 to Motif 10, and their lengths ranged from 28 to 50 amino acid residues (Figure 4). Each gene member contains 8 to 10 conserved motifs. All the motifs except Motifs 7 and 9 are included in CqVOZ1 and CqVOZ3. However, CqVOZ2 and CqVOZ4 contain all the ten motifs. To further analyze the conserved properties of the quinoa VOZ transcription factor family, multiple amino acid sequence alignments of the VOZ domains were performed (Figure 5). The results indicated that the VOZ domain is highly conserved and formed from the full sequences of Motif 2, 3, 1, 4, and 6. The functions of most of the conserved motifs remain to be clarified. They may influence the activation or inhibition of VOZ transcription factors of quinoa by binding or interacting with other genes. Generally, the conserved motif compositions and similar gene structures of the VOZs in the same group, combined with phylogenetic analysis results, strongly support the reliability of the group classifications. 

### 3.4. Analysis of CqVOZ Protein Structures

The secondary structure of CqVOZ proteins was analyzed, and the prediction results are shown in Figure 6A. Similarly, all the CqVOZ proteins were mainly composed of an α helix, extended strand, and random coil, of which random coil accounted for the largest proportion with an average of 58.75%, followed by α helix with an average of 27.81%, and extended strand accounted for a relatively small proportion with an average of 13,44% (Appendix A). To obtain the reasonable theoretical structures of CqVOZ proteins, they were modeled by a SWISS-MODEL server. The predicted 3D structures are shown in Figure 6B. According to structure similarity, CqVOZ1 and CqVOZ3 can be grouped together, while CqVOZ2 and CqVOZ4 are in a group, consistent with phylogenetic tree grouping results. All CqVOZ proteins were mainly random coil elements, basically similar to the prediction results of the secondary structure, indicating its credibility.

### 3.5. CqVOZs Expression Profiling in Different Tissues

Analysis of tissue-specific gene expression profiles can provide evidence for revealing the potential functions of genes throughout development. In this study, qRT-PCR was performed to investigate the expression variation of the *CqVOZ* gene family in different tissues and development stages of quinoa, including root, stem, leaf, inflorescence, and seeds 10 days after pollination (seed-10 DAP), seed-20 DAP, and seed-30 DAP. The results indicated that the *CqVOZ* genes had obvious tissue expression specificity (Figure 7). All four *CqVOZs* exhibited higher expression in inflorescence and lower expression in stem and leaf, suggesting these genes may be closely involved in flower development. In detail, *CqVOZ1* and *CqVOZ4* were highly expressed in inflorescence, followed by root, and constitutively expressed in seeds of different developmental days in a low expression manner. *CqVOZ2* showed increased expression during seed development, but the expression levels were generally lower than in inflorescence. *CqVOZ3* was highly expressed in seeds of different stages and inflorescence, demonstrating that it might play an important role in the formation of grain yield of quinoa. 

### 3.6. Expression Analysis of CqVOZs under Abiotic Stress Treatments

As a multifunctional factor, *VOZ* has also been reported to play critical roles in responding to biotic and abiotic stresses. To further explore the potential functions in resistance to various abiotic stresses, the gene expression patterns of *CqVOZs* under cold, salt, and drought stress treatments were evaluated (Figure 8). Under cold stress, *CqVOZ2* and *CqVOZ3* were significantly induced and the expression levels peaked at 24 h, up to 6 and 18 fold, respectively (Figure 8A). The relative expression levels of *CqVOZ1* and *CqVOZ4* were slightly affected by low-temperature treatment, with *CqVOZ1* slightly decreased and *CqVOZ3* slightly increased, but not significantly (Figure 8A). Under salt stress, compared with the control group, *CqVOZ2* and *CqVOZ3* were significantly up-regulated, while *CqVOZ1* and *CqVOZ4* were down-regulated at each time point (Figure 8B). *CqVOZ2* and *CqVOZ3* had their highest expression levels at 12 h and 3 h, respectively, but the expression of *CqVOZ3* increased significantly more than that of *CqVOZ2*, implying that *CqVOZ3* could respond more quickly to salt stress. Under drought stress, the expressions of *CqVOZ1* and *CqVOZ4* were significantly inhibited, while *CqVOZ2* and *CqVOZ3* increased gradually over time until peak expression at 24 h and 12 h, respectively, up to 3 and 6 fold higher than control (Figure 8C). In summary, the expressions of the *CqVOZ* gene family were induced or repressed under various abiotic stress conditions. *CqVOZ1* and *CqVOZ4* were negatively responsive to salt stress and drought stress, while another two genes responded positively to cold stress, salt stress, and drought stress. 

## 4. Discussion

Plants are subjected to various biotic and abiotic stresses during growth and development, which could dramatically decrease crop yield and quality and even cause death. As key molecular switches regulating stress-responsive gene expression, transcription factors are important for functioning in these adverse environments. Vascular plant one zinc-finger proteins are plant-specific transcription factors, and this gene family plays vital roles in the regulation of plant growth and development and VOZ-mediated abiotic stress response. *VOZ* was first discovered while studying genes associated with flowering time in Arabidopsis [14], and since then, the *VOZ* gene family has been extensively studied in many plants. However, as one of the most promising crops in the future, quinoa has high nutritional and stress-tolerant properties, but no comprehensive analysis of the *VOZ* gene family and their functions in growth and development and abiotic stress response has been reported yet. The completion of quinoa genome sequencing and the in-depth study of transcriptomics will help to discover resistance genes and promote the genetic improvement of quinoa [38,46]. In the current study, for the first time, the *VOZ* gene family was identified from the quinoa genome, and differences and variations in gene structure, conserved motifs, and protein structure were also detected. The analysis of expression data further revealed that *VOZ* genes are potential participants in the regulation of quinoa growth and development and stress resistance. The genome-wide identification of members of the VOZ transcription factor family in quinoa is an important starting point for further research on their biological functions and provides some information for studying *VOZ* genes in other species. 

VOZ transcription factor family is a small family with 635 members and their homologous in 166 plant species [47]. In the present study, a total of 4 *VOZ* genes were identified from the quinoa genome database using bioinformatics methods and named *CqVOZ1* to *CqVOZ4* on the basis of their chromosomal location. The size of the *VOZ* gene family in quinoa was larger than that of sorghum, sugarbeet, and model plants Arabidopsis and rice, all of which contained two members, but lower than that of corn (five members), soybean (six members), and cotton (seven members) (Figure 2). It has been reported that the abundance of transcription factors largely depends on gene sequence duplications during genome evolution [48]. The differences in the number of *VOZ* genes among different plant species may be due to the occurrence of genomic duplication events, resulting in the amplification of loss of *VOZ* genes in the allotetraploid plant quinoa. 

An accurate evolutionary history is the first step to understanding the evolutionary relationships of genes and may provide some valuable information for studying the biological functions of genes and the pathways involved in or regulated by genes [49]. To determine the evolutionary relationships of VOZ proteins, a phylogenetic tree based on the amino acid sequences of VOZs in quinoa, Arabidopsis, rice, sorghum, sugarbeet, corn, cotton, and soybean was constructed. The results indicated that all the VOZ proteins were divided into four groups, four CqVOZs located in the nucleus belonged to Group II and III. Quinoa VOZ proteins were clustered with dicotyledons Arabidopsis, sugarbeet, cotton, and soybean, but not with monocotyledons rice, sorghum, and corn, suggesting that the differentiation of VOZ transcription factors occurred after the divergence of monocotyledons and dicotyledons. Moreover, quinoa CqVOZs showed the highest homology with VOZs of the same family plants, indicating the accuracy of the botanical classification. The results above conform to the current understanding of plant evolutionary history [50]. 

The exon/intron structural diversity plays a crucial role in the evolution of plant gene families [51,52,53]. There are three main mechanisms for the differences in exon/intron structure, gain or loss, exonization or pseudoexonization, and insertion or deletion. In this work, the number of introns in *CqVOZ* genes is slightly different, except for the *CqVOZ2* gene which contains only two introns, the other three genes all have four introns (Figure 3), demonstrating that essential differentiations have arisen during the evolution of this gene family in quinoa. However, some studies have revealed that the fewer the gene introns in plants, the stronger the adaptability to the external environment [54]. Hence, as a gene with fewer introns, *CqVOZ2* probably plays an important role in response to various biotic and abiotic stimuli. 

Different VOZ family members contained highly conserved domains, which might be closely related to their identical and similar regulatory functions [26]. In this study, 10 conserved motifs were also identified to further study the evolution of quinoa CqVOZ proteins. The results showed that each member contained at least eight conserved motifs, all the 10 motifs were included in CqVOZ2 and CqVOZ4, while CqVOZ1 and CqVOZ3 contained only eight of these motifs (Figure 4). To some extent, it also confirmed the reliability of subfamily classification. On the other hand, the different motif compositions could result in functional diversity [55]. Multiple sequence alignments demonstrated that the VOZ domain was highly conserved, sharing common motifs 1, 2, 3, 4, and 6 (Figure 5). All of these suggested that the *VOZ* family genes in quinoa are relatively conserved for evolution. Additionally, to further understand the protein structure characteristics of CqVOZs, the 3D structures were constructed. It indicated that the protein structural elements were mainly random coils. CqVOZ1 and CqVOZ3 had high similarity in the spatial structure, while CqVOZ2 and CqVOZ4 seemed more similar (Figure 6), which was consistent with the phylogenetic tree grouping results and verified its reliability to some extent. 

The analysis of gene expression profiles can provide useful clues for function prediction. In this study, the *VOZ* gene expression patterns in different quinoa tissues were reported. The results showed that four *CqVOZs* displayed significant tissue expression specificity, indicating their diverse roles in the growth and development of quinoa. As described, there are important functions of *AtVOZ* family genes in regulating flowering in Arabidopsis [19]. We also found that all the *CqVOZs* were expressed at relatively higher levels in inflorescence than those in other quinoa tissues (Figure 7), suggesting they might be critical participants in floral development. Grain formation is a key trait in current and future crop high-yield breeding, and the analysis of the genetic regulatory network is an important basis for improving this trait. Here, *CqVOZ2* and *CqVOZ3* were found to be highly expressed during seed development, which might serve as valuable genes for improving quinoa yield. It has been demonstrated that *VOZ* genes also play key roles in responding to abiotic and biotic stresses [22,23]. With the goal of identifying candidate abiotic stress-responsive *CqVOZ* genes, qRT-PCR was conducted to analyze the expression profiles of quinoa *VOZ* family genes under various abiotic stresses (Figure 8). *CqVOZ2* and *CqVOZ3* were induced by cold, salinity, and drought stresses, and showed significantly up-regulated expression patterns. These two genes can be selected as key candidate genes for further functional research on abiotic stress resistance. On the contrary, the expression level of *CqVOZ1* was inhibited by cold, salinity, and drought stresses, which might be considered a negative regulator in response to abiotic stresses. *CqVOZ4* was positively regulated in response to cold stress but negatively regulated under salinity and drought stresses, implying that this gene might play different roles in resistance to different abiotic stresses. Hence, different *CqVOZs* had different responses and regulatory mechanisms under different treatments. Furthermore, the *CqVOZ* gene family is of great significance for quinoa to adapt to the complicated and changeable soil and climate environment. However, further experiments are required to verify their biological functions and reveal the signaling pathways they may be involved in.

## 5. Conclusions

In this study, a total of four *CqVOZ* genes were identified in the quinoa genome using bioinformatics methods. These genes are unequally distributed on three chromosomes and were phylogenetically divided into two subfamilies. The basic characteristics, genome location, gene structures, conserved motifs, multiple sequence alignment of conserved domains, and secondary and tertiary structures of these genes were analyzed in detail, providing a solid foundation for understanding the evolutionary relationships of the *CqVOZ* gene family. Gene expression patterns were analyzed using qRT-PCR, which indicated that *CqVOZs* played important roles in quinoa growth and development as revealed by their expression profiles in different tissues and in response to various abiotic stresses. These results provide a basis for further study of the biological roles of the VOZ family in quinoa and have important practical significance for quinoa resistance breeding.

## Figures and Tables

**Figure 1 genes-13-01695-f001:**
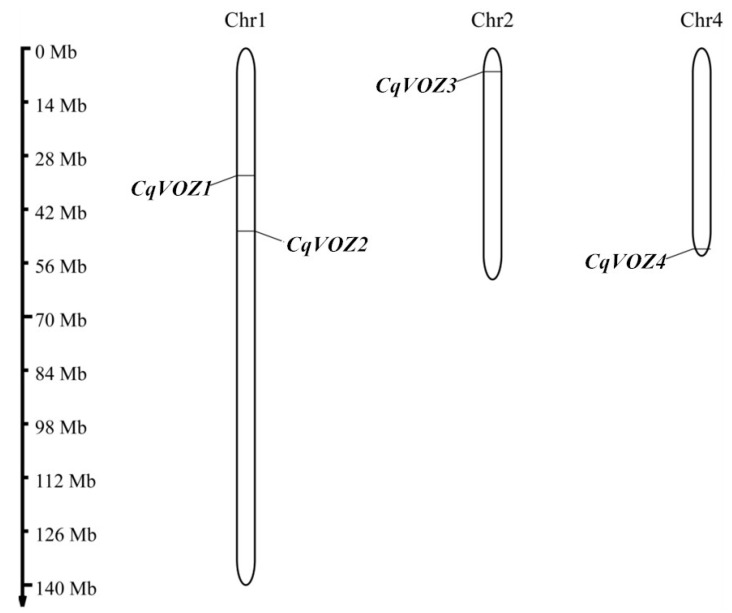
Chromosomal distribution of *CqVOZ* genes.

**Figure 2 genes-13-01695-f002:**
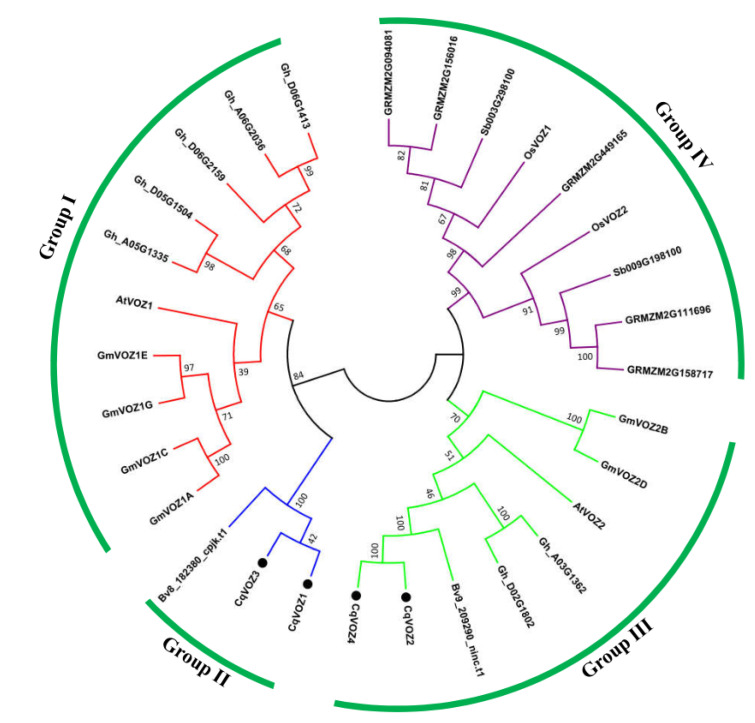
Phylogenetic relationships of the VOZ transcription factors from quinoa, Arabidopsis, rice, cotton, sorghum, corn, soybean, and sugarbeet. A neighbor-Joining tree was constructed using MEGA 5.0 with 1000 bootstrap replicates. *CqVOZs* were specifically marked by black circles.

**Figure 3 genes-13-01695-f003:**
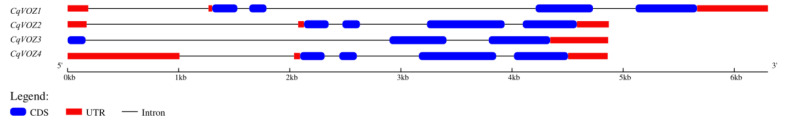
Exon-intron structures of the *CqVOZ* genes. The blue boxes represent the exons. The red boxes represent the 5′ UTR and the 3′ UTR. Lines represent the introns.

**Figure 4 genes-13-01695-f004:**
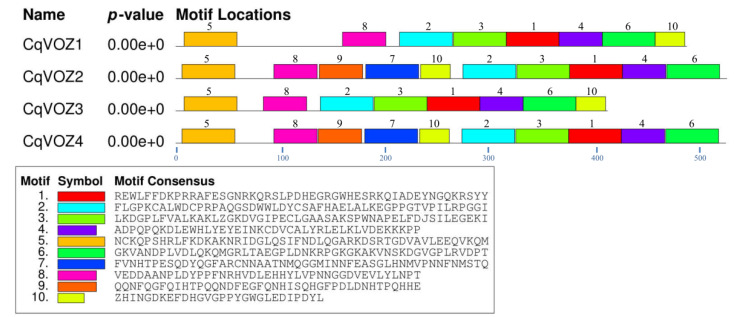
The conserved motifs of CqVOZ proteins. Distribution of these motifs was identified by MEME, and boxes of different colors represent different motifs. The consensus sequences and the amino acid lengths of these motifs were also listed.

**Figure 5 genes-13-01695-f005:**
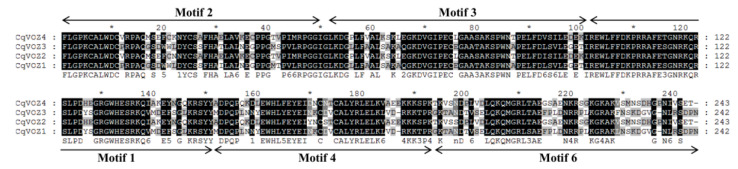
Amino acid sequence alignments of the VOZ domains of CqVOZs. Names of all *CqVOZ* genes are shown on the left side of the figure. The motifs 2, 3, 1, 4, and 6 formed the putative VOZ domain.

**Figure 6 genes-13-01695-f006:**
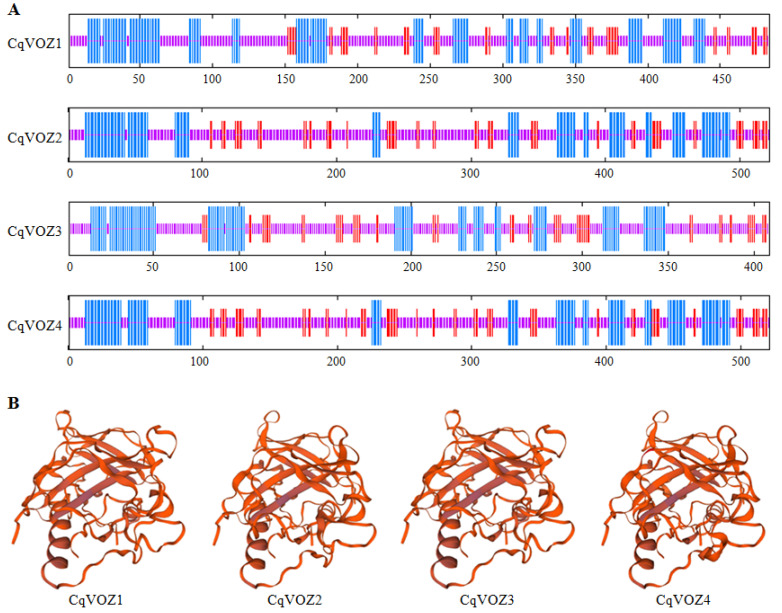
Structural analysis of four CqVOZ modeled proteins. (**A**) Secondary structure of CqVOZ proteins. α helix is indicated in blue, extended strand is indicated in red, random coil is represented in purple. (**B**) 3D structure of CqVOZ proteins.

**Figure 7 genes-13-01695-f007:**
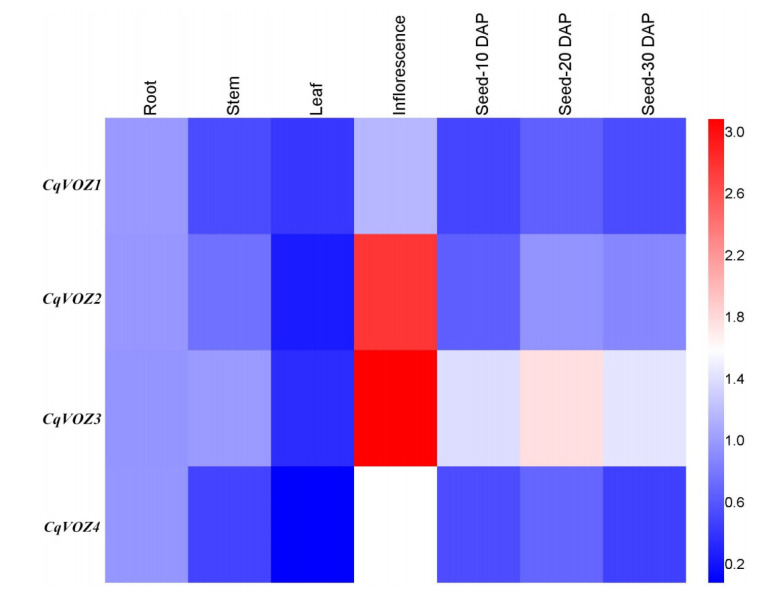
Expression profile of *CqVOZ* gene family in different tissues and developmental stages. The X-axis represents samples; Y-axis represents *CqVOZ* gene names. The red and blue colors represent high and low values of relative expression levels according to qRT-PCR, respectively.

**Figure 8 genes-13-01695-f008:**
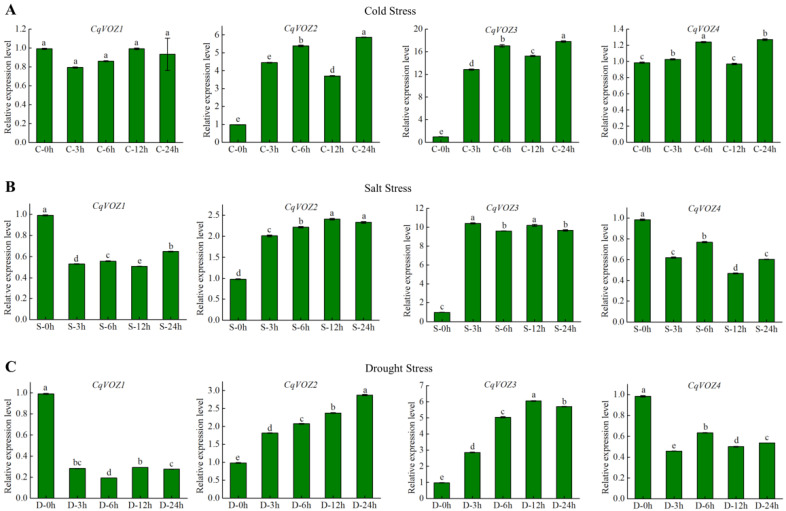
Expression analysis of *CqVOZs* under cold (**A**), salt (**B**), and drought (**C**) stress treatments. The Y-axis represents the relative expression levels, and the X-axis shows different time points of stress treatments. Bars indicate the mean values of three biological replicates ± standard deviation. Columns with different letters represent significant differences at *p* < 0.05 level (Duncan’s test).

**Table 1 genes-13-01695-t001:** The sequences of primers used for qRT-PCR.

Gene Name	Forward Primer Sequence (5’–3’)	Reverse Primer Sequence (5’–3’)
*CqEF1α*	GTACGCATGGGTGCTTGACAAACTC	ATCAGCCTGGGAGGTACCAGTAAT
*CqVOZ1*	GCTGTAACTTCAGATATTTGAAG	GTGCTGGGACATTGTGCACAT
*CqVOZ2*	CTCCTCCAATCTAGGGCATCT	CTCGGAATTAGAGACAGAATCG
*CqVOZ3*	CTCTCTATCGTTGTCATCATCT	GTTGATAAGATCTTTGTTATAA
*CqVOZ4*	GTCGACCGAGTAGGAGAAGA	CCTCGTACTAAGTCACCACATTAC

**Table 2 genes-13-01695-t002:** The information of *CqVOZ* genes.

Gene Name	Gene ID	Chr	Genomic Location	ORF	Exon	AA	MW (kDa)	pI	Subcellular Localization
*CqVOZ1*	AUR62014889	1	32490238..32496540	1455	4	484	54.31	5.70	Nucleus
*CqVOZ2*	AUR62024758	1	46806664..46811534	1569	4	522	58.08	5.48	Nucleus
*CqVOZ3*	AUR62031095	2	5865078..5869942	1230	3	409	45.93	7.56	Nucleus
*CqVOZ4*	AUR62037692	4	51222056..51226916	1566	4	521	57.93	5.49	Nucleus

## Data Availability

Not applicable.

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
