# Peer review of "Genome-Wide Analysis and Expression Profiles of the VOZ Gene Family in Quinoa (Chenopodium quinoa)"

_genes, 2022, doi:10.3390/genes13101695_

Round 1

Reviewer 1 Report

The article entitled, “Genome-Wide Analysis and Expression Profiles of the VOZ Gene Family in Quinoa” sum up the genome-wide analysis of the VOZ Gene. It is a regular analysis. Author needs to perform a few more analysis before acceptance of this Ms. Further, I could see several grammatical/typographical errors.

  1. How did the author confirm the number of VOZ genes in quinoa species?
  2. The authors are suggested to perform the domain analysis in the VOZ sequences.
  3. The authors should explain the significance of phylogenetic analysis.
  4. Protein protein interaction analysis and miRNA interaction analysis  should be the part of Ms. Author may refer the suggested Ms for the analysis. https://www.mdpi.com/2075-1729/12/7/941, https://www.mdpi.com/2223-7747/11/5/587
  5. Evolutionary analysis using Ka/Ks should also be included https://www.mdpi.com/2223-7747/11/7/911

Suggested minor revision

  1. There are many grammatical and typographical errors throughout the manuscript which need to be corrected after thorough reading. The manuscript is very poorly written.
  2. Few of the lines mentioned below should be checked and corrected-

·                  Line no. 56: spacing between words.

·                  Line no. 94: spacing between words.

·                  Line no. 96: The year instead of “And the year”.

·                  Line no. 339: Sentence is not clear, grammar correction required.

·                  Line no. 369: “Moreover, quinoa CqVOZs were more closely related to those of the same plant family than to other species, indicating the accuracy of plant classification”, the line needs to be corrected to make it more understandable. This line is not explanatory.

Author Response

  1. How did the author confirm the number of VOZ genes in quinoa species?

Response: The conserved amino acid sequences of AtVOZ and OsVOZ proteins were used as queries to perform BlastP alignment in quinoa genome, and the proteins without VOZ domain and incomplete coding were removed, at last we obtained all the four CqVOZ transcription factor family members.

  1. The authors are suggested to perform the domain analysis in the VOZ sequences.

Response: Amino acid sequence alignment analysis of the VOZ domain has been performed in Figure 5 in this article.

  1. The authors should explain the significance of phylogenetic analysis.

Response: “We found that the CqVOZ genes were more closely related to genes from sugarbeet, cotton, Arabidopsis and soybean than to genes from rice, sorghum and corn.”

  1. Protein protein interaction analysis and miRNA interaction analysis  should be the part of Ms. Author may refer the suggested Ms for the analysis. https://www.mdpi.com/20751729/12/7/941,https://www.mdpi.com/2223-7747/11/5/587

Response: Thanks for your suggestion but protein protein interaction analysis and miRNA interaction analysis are technically difficult, which will be considered in subsequent research. 

  1. Evolutionary analysis using Ka/Ks should also be included https://www.mdpi.com/2223-7747/11/7/911

Response: Ka/Ks analysis has important applications in the study of molecular evolution of nucleic acids. However, the focus of this study is not on evolutionary analysis but on the transcriptional response of CqVOZs.

Suggested minor revision

  1. There are many grammatical and typographical errors throughout the manuscript which need to be corrected after thorough reading. The manuscript is very poorly written.

Response: The grammatical and typographical errors throughout the manuscript have been corrected after through reading.

  1. Few of the lines mentioned below should be checked and corrected-

Line no. 56: spacing between words.

Line no. 94: spacing between words.

Line no. 96: The year instead of “And the year”.

Line no. 339: Sentence is not clear, grammar correction required.

Line no. 369: “Moreover, quinoa CqVOZs were more closely related to those of the same plant family than to other species, indicating the accuracy of plant classification”, the line needs to be corrected to make it more understandable. This line is not explanatory.

Response: Line no. 56, Line no. 94 and Line no. 96 had been modified as required. The sentence in Line no. 339 was changed as “The completion of quinoa genome sequencing and the in-depth study of transcriptomics will help to discover resistance genes and promote genetic improvement of quinoa.” The sentence in Line no. 369 was changed as “Moreover, quinoa CqVOZs showed the highest homology with VOZs of the same family plants, indicating the accuracy of the botanical classification.”

Reviewer 2 Report

The authors have carried out genome-wide analysis and expression profiles of the VOZ2 gene family in Quinoa plant. The work is original and interesting, however I have few questions/ suggestions which need to be incorporated by the authors.

1. Why the authors used PEG6000 for drought stress and not PEG8000? PEG6000 can enter the cells and cause osmotic damage. Therefore, high molecular weight PEG like PEG-8000 is recommended for inducing drought stress. 

2. In figure 8, details of statistical analysis are missing. Was the data statistically analyzed? If so, mention the details in figure legend. Also show significant differences on bar graph through different alphabets or asteriks.  

3. Add reference for for 2−∆∆CT method in materials and methods.

4. Add your data figure's citation in the discussion part.

Author Response

  1. Why the authors used PEG6000 for drought stress and not PEG8000? PEG6000 can enter the cells and cause osmotic damage. Therefore, high molecular weight PEG like PEG-8000 is recommended for inducing drought stress. 

Response: PEG6000 has been extensively used for drought stress in many valuable researches, and PEG6000 will not cross the cell wall, with stable chemical properties, less toxic to plants, suitable for inducing drought stress.

  1. In figure 8, details of statistical analysis are missing. Was the data statistically analyzed? If so, mention the details in figure legend. Also show significant differences on bar graph through different alphabets or asteriks.  

Response: Yes, the data was statistically analyzed and the details were mentioned in figure legend. A new figure 8 showed significant differences on bar graph through different alphabets.

  1. Add reference for 2−∆∆CTmethod in materials and methods.

Response: I have added reference for  2−∆∆CT method in materials and methods.

  1. Add your data figure's citation in the discussion part.

Response: The data figure’s citation has been added in the discussion part at the corresponding position.

Reviewer 3 Report

The authors have well characterized the VOZ gene family in tree spp. However only four candidates are identified in the study but phylogeny with other model plants will help to understand the importance of their functional diversity. I have some queries as bellow. 

1. The authors have performed RT analysis in abiotic stresses. But effect of these abiotic stresses on studied plant species are missing in introduction. Please prepare the proper background for the study. 

2. The functional annotation through GO analysis will also provide useful information, which is found missing. 

3. May places the name of the genes is not in italics. Correct it in throughout manuscript. 

4. The authors have analyzed only the early interaction to abiotic stresses and expression of VOZ candidates. If they perfomed at different days, it will be more useful. 

5. The authors analyzed the expression pattern in roots under aniotic stresses. For the salt and drought its ok to analysze in the roots. But for clod (short period only upt 24h) first affect the leaves followed by roots.  

6. The discussion portion need to be improved. 

7. I also markerd the revisions in PDF manuscript. 

Author Response

  1. The authors have performed RT analysis in abiotic stresses. But effect of these abiotic stresses on studied plant species are missing in introduction. Please prepare the proper background for the study. 

Response: The content of the effect of abiotic stresses on quinoa has been added in introduction portion and the references are also added.

  1. The functional annotation through GO analysis will also provide useful information, which is found missing. 

Response: Thanks for your suggestion. GO analysis mainly aims to study the functions of the identified differentially expressed genes, and this is what we are going to do in subsequent transcriptome studies.

  1. May places the name of the genes is not in italics. Correct it in throughout manuscript. 

Response: The gene names are corrected as italics and the protein or transcription factor names still remain not in italics throughout the manuscript.

  1. The authors have analyzed only the early interaction to abiotic stresses and expression of VOZ candidates. If they perfomed at different days, it will be more useful. 

Response: The transcription response of transcription factors under abiotic stresses is rapid and the expression is transient. If the stress treatments are performed at different days, the expression of VOZ candidates is easy to recover.

  1. The authors analyzed the expression pattern in roots under abiotic stresses. For the salt and drought its ok to analysze in the roots. But for clod (short period only upt 24h) first affect the leaves followed by roots. 

Response: Thanks for your suggestion and that is what I will pay attention to in subsequent research. Maybe it is better to take leaves rather than roots to analyze gene expression pattern under cold stress, but  in many valuable studies, researchers also select roots for expression analysis.

  1. The discussion portion need to be improved. 

Response: Many grammatical and typographical errors have been corrected in the discussion portion.

  1. I also markerd the revisions in PDF manuscript. 

Response: It has been modified as suggested.

Reviewer 4 Report

Dear authors,

I am really eager for your research. That was good but there were some problems in the references section. Please make references based on journal format.

Author Response

I am really eager for your research. That was good but there were some problems in the references section. Please make references based on journal format.

Response: All the references have been corrected based on journal format in the references section.

Round 2

Reviewer 1 Report

The Ms is mostly in-silico work except Rtpcr, that too with very basic analysis. All such data are easily available at ensemble plants. They should do indepth evolutionary analysis and interaction analysis to strengthen the work.

Authors have ignored all the suggested analysis.

Author Response

The Ms is mostly in-silico work except Rtpcr, that too with very basic analysis. All such data are easily available at ensemble plants. They should do indepth evolutionary analysis and interaction analysis to strengthen the work.

Authors have ignored all the suggested analysis.

Response: The focus of this study is to mine candidate resistance VOZ genes in quinoa at the genome-wide level. It is found that protein protein interactions and Ka/Ks were also not analyzed in some references for gene family analysis. doi: 10.3390/genes12121867, doi: 10.3390/genes11091032. And we will focus on the verification of VOZ gene function and analysis of interacting proteins in subsequent research.

Reviewer 3 Report

Now the manuscript looks fine. Biut still some minor corrections are required. 

1. Pls add the scientific name of the plant in trhe title. 

2. At many place in the text, the name of genes not italic, pls correct it. 

3. The Pfam ID of VOZ domain may be added in the text, either in materials and methods or in the text for readers. 

4. All the scientific names should be in italics, please correct through out manuscript. 

5. In figure 8; i think its repeated. pls check it. The y axis can be made uniform as one decimal point. 

6. I am also attched here the manuscript. I also hoghlighted the revisions in the text. 

Author Response

  1. Pls add the scientific name of the plant in the title. 

Response:Chenopodium quinoa” has been added in the title.

  1. At many place in the text, the name of genes not italic, pls correct it. 

Response: Gene names have been corrected to italics, and protein names are still non-italics.

  1. The Pfam ID of VOZ domain may be added in the text, either in materials and methods or in the text for readers. 

Response: “CqVOZ protein sequences were submitted to Pfam (http://pfam.xfam.org) to identify conserved VOZ domains” is added in materials and methods portion.

  1. All the scientific names should be in italics, please correct through out manuscript. 

Response: All the scientific names have been corrected to italics through out manuscript.

  1. In figure 8; i think its repeated. pls check it. The y axis can be made uniform as one decimal point. 

Response: The figure without significance analysis is deleted. I tried setting the y axis to one decimal point, but when these numbers in y axis are integers, Origin 8 software defaults to no decimal point.

  1. I am also attached here the manuscript. I also highlighted the revisions in the text. 

Response: All the places have been corrected as suggested.